# Information Theory in Perception of Form: From Gestalt to Algorithmic Complexity

**DOI:** 10.3390/e27040434

**Published:** 2025-04-17

**Authors:** Daniel Algom, Daniel Fitousi

**Affiliations:** 1School of Psychological Sciences, Tel-Aviv University, Tel Aviv-Yafo 6997801, Israel; 2School of Communication Disorders, Achva Academic College, Arugot 4567889, Israel; 3Department of Psychology, Ariel University, Ariel 4076414, Israel; danielfi@ariel.ac.il

**Keywords:** gestalt, information, sets, redundancy, algorithmic complexity

## Abstract

In 1948, Claude Shannon published a revolutionary paper on communication and information in engineering, one that made its way into the psychology of perception and changed it for good. However, the path to truly successful applications to psychology has been slow and bumpy. In this article, we present a readable account of that path, explaining the early difficulties as well as the creative solutions offered. The latter include Garner’s theory of sets and redundancy as well as mathematical group theory. These solutions, in turn, enabled rigorous objective definitions to the hitherto subjective Gestalt concepts of figural goodness, order, randomness, and predictability. More recent developments enabled the definition of, in an exact mathematical sense, the key notion of complexity. In this article, we demonstrate, for the first time, the presence of the association between people’s subjective impression of figural goodness and the pattern’s objective complexity. The more attractive the pattern appears to perception, the less complex it is and the smaller the set of subjectively similar patterns.

## 1. Shannon’s Contribution

Information is often the vehicle for our perceptions. Its formal development by Shannon (1948) gave new impetus to the study of perception [1]. In particular, it provided a potentially powerful tool for rigorous definitions of key Gestalts terms like figural goodness, pattern-ness, or randomness. Once a crucial stumbling block was removed, the application of information theory opened up a wealth of exciting avenues for research. Our goal in this report is to follow these developments up to their recent culmination in applications of algorithmic complexity to Gestalt stimuli. 

Claude Shannon’s publication of a mathematical theory of communication [1] revolutionized how cognitive psychologists approach their subject. Nowhere was the impact more powerful than in the field of visual perception. Half of a century before Shannon’s contribution, Gestalt psychologists had rekindled interest in the Greek theme of symmetry [2], introducing the notion of “figural goodness” [3,4]. Good figures are simple, symmetric, regular, orderly, and predictable. The problem was that these compelling impressions were notoriously subjective and lacked rigorous definitions and assays. Experimental methods were needed to quantify the determinants of figural goodness. These tools were provided by Shannon’s epoch-making article (see also [5]). Since the 1950s, hundreds of experiments have been conducted within the framework of “information theory” (e.g., [6]). Exciting new ideas emerged as a result of that effort (e.g., [7,8]).

The superfecundity of ideas granted, the application of information theory in psychology has been fraught with problems. A major one is associated with the notion of set (of alternatives) in information theory. Grappling with the problem was difficult. However, the added bonus was valuable discoveries that enriched cognitive science. This is the topic of the present study.

## 2. The Problem of Psychological Sets

We assume a rudimentary familiarity with Shannon’s statement of information. Here, we provide only the basic definition to highlight the problem of set for applications in psychology. Let x_i_ be an event from a set of N alternative events with a probability of p(x_i_). Then, the information *I* carried by the occurrence of x_i_ can be defined as,*I*(x_i_) = −log p(x_i_) in bits(1)
with the bit, the unit of information, signalling that the logarithm refers to the base of 2. If each alternative event in the set has the same probability of occurrence, then one obtains the following:*I*(x) = log N (2)

We present Equation (2) at the cost of restating the triviality that without the presence of a set of N alternatives, there is no information to any single stimulus x. It might be unsettling to realize that N was missing from the early attempts of applying information theory to visual perception. We hasten to add that outside visual perception of form, N did not pose a problem. When examining transmitted information, the goal of most early studies, the stimulus set is given by the experimenter so that the question is focused on the amount of information transmitted between the input and the output (e.g., [9,10,11]. In sharp contrast, the studies of form scrutinized *single* stimuli (e.g., the picture of a cat, the drawing of a tree or of a building). This style of studies rapidly hit an impasse due to the missing notion of N in the analysis (e.g., [12,13,14,15]). However, the notion of set, of N, is indispensable.

Considering Equations (1) and (2) again, applications of information theory *mandate* considering each stimulus x_i_ as a member of a set of alternatives N. Consequently, there exists a set of alternatives whenever one perceives a stimulus. When a person perceives the book on the desk, the set of alternative stimuli in which the book is a member is activated ineluctably. The challenge is to identify the operative set with each act of perception. The challenge seemed insurmountable until the solution proposed by Wendell R. Garner [16,17,18], thereby laying the foundations for a genuine psychological information theory.

## 3. Garner’s Solution: Sets Varying in Information Content

Despite the difficulty and apparent defiance of common sense (and probably against the inner resistance of many a psychologist), Garner did not shy away from the inescapable deduction from information theory: there is no such thing as perception of a single stimulus. Closely following Shannon’s development, Garner maintained that “The single stimulus has no meaning except in the context of alternatives…. the organism infers sets of stimulus alternatives, and without these inferred sets, no one can describe the single stimulus” ([18], p. 185). In other words, the perception of a stimulus depends on its set of alternative stimuli (the N in Equation (2))—those stimuli that could have appeared but did not in that case. The stimulus derives its meaning from that set; the set shows what the stimulus is not (and what it might have been). Again, the challenge is to retrieve the operative set.

In everyday life, the context readily affords the set of alternatives. When you pick *The turn of the screw* from your shelf, the likely set is the other books by Henry James that you own. Presented with the numeral 7, the set may include the single digits 1–9, or the neighbors, i.e., 6 and 8. Responding to the letter k, the set may well be the letters of the alphabet or the subset of consonants. The problem becomes formidable (insoluble?) when perceiving a meaningless stimulus. What is the set of alternatives for a dot pattern? For an irregular form? For a triangle? Garner’s groundbreaking idea was subjecting the stimulus to spatial transformations so that the group of transformations comprise the sought set [17,18]. 

Consider the pair of patterns in Figure 1. Both comprise five dots placed in the imagery cells of a 3 × 3 matrix. These were the stimuli used for demonstrating the spatial transformation idea. Each dot pattern can be rotated and reflexed in 90° steps around the vertical, horizontal, left, and right diagonal, and each new pattern thus produced is a member of the rotation and reflection (R&R) subset of alternatives. Inspecting the patterns of Figure 1, the pattern on the left has an R&R subset of 1—the pattern remains the same throughout all rotations and reflections. The pattern at the right of Figure 1 has an R&R subset of 8—the number of new patterns emerging with each rotation and reflection.

Readers will readily recognize that the R&R subset idea is isomorphic with the notion of symmetry in mathematical group theory. Thus, the symmetry group of the pattern at the left is 8, whereas that on the right is 1. Garner’s theory concerns sets and members of sets, whereas group theory is concerned with transformations and the type of each transformation. One measure is the reverse of the other [19]. In mathematics, “We say something is symmetrical, when it looks the same from more than one point of view” ([20], p. 25). This perfectly aligns with Garner’s theory.

Once it is recognized that a single stimulus has meaning only as a member of a set, the information theory notion of *redundancy* becomes relevant. One can say that the pattern on the left is redundant (since each member of the R&R subset is a replication of all other members). The pattern on the right, by contrast, is not redundant and carries information. In general, the smaller its R&R subset, the more redundant and less informative is the stimulus. The necessary relationship between sample size and redundancy was best demonstrated by Garner [16,18], and it is shown next.

## 4. Subset Size and Redundancy

The relation is succinctly stated as follows by Garner [16]: “selection of fewer stimuli than can be generated with a given number of variables produces a redundant set of stimuli” (p. 321). Think of a set of stimuli produced by crossing the values of the constituent variables. Then, any sample or subset drawn from this total set *must* contain correlation between the constituent variables. The theorem is best understood through the visual illustration by Algom and Fitousi [21], which, in turn, was inspired by [18]. The illustration is presented in Figure 2.

The total set shown in Figure 2 is created by crossing all the binary values of three variables: global form (circle, square), slope of the diagonal (upwards, downwards), and type of line (solid, broken). Note that all combinations are realized in this total set of the eight stimuli. Now consider possible samples or subsets drawn from this total set. Discarding all the circles will not do, since we are left then with a total set defined by two variables. An example of an actual subset is given in Figure 3.

The correlation between the variables in the subset is not accidental; all the subsets *must* contain correlation or redundancy, and the smaller the subset, the larger redundancy. In the extreme, the smallest subsets drawn entail a very large amount of redundancy so that they convey virtually no information.

## 5. Figural Goodness and Set Size

Thus far, our discussion has been technical in nature, identifying the constraints that should be obeyed in proper applications of information theory to psychology. We now complement the discussion by considering several psychological phenomena uncovered in such applications. Consider Figure 1 again. The pattern at the left comprises a good Gestalt—it is simple, balanced, symmetric, orderly, predictable, and pleasing to the senses. This harkens back to antiquity; in the Greek word, symmetry meant harmony, balance, and beauty [2,22]. The pattern at the right lacks in these properties. Discussing the question of how Gestalts are created in the first place would take us too afield. For early processes of creating Gestalts (e.g., grouping, segregation), one may consult the research by Kimchi and her associates [23,24,25,26]. 

In a classic study, Garner and Clement [27] attempted to capture the former subjective terms by providing objective empirical measures. They created 90 dot patterns like those in Figure 1 and asked the participants to perform two separate tasks: (a) rating each pattern for figural goodness and (b) placing each pattern in a group with other patterns based on similarity. The groups need not be of equal size. The most remarkable result of the study was the correlation found between (a) subjective figural goodness and (b) the size of the groups created based on similarity. Good figures tended to reside in small groups of similarity. The very best figures (the pattern at the left in Figure 1) were literally peerless or groupless—the size of its group was 1. Garner’s [17] title captures the main takeaway, “Good patterns have few alternatives”. The summary of that paper is equally instructive, to wit, “Poor patterns are those which are not redundant and thus have many alternatives, good patterns are those which are redundant and thus have few alternatives, and the very best patterns are those which are unique, having no perceptual alternatives” (p. 42).

Let us pause briefly to reconnect the Garner and Clement [27] results with the previous development. We should be careful to distinguish between subjective and objective variables. Thus, the size of an *R&R subset* or size of a *symmetry group* are objective variables rigorously defined. In contrast, the participant-created *similarity group* (e.g., by sorting cards) and, certainly, the rating of *figural goodness* are subjective variables. Now, there is a close monotonic relation between the size of *R&R subsets*, on the one hand, and the size of subjectively created similarity groups, on the other hand. One can say that the former is the objective counterpart of the latter, and both the subjective and the objective groups relate to the perception of figural goodness the same way: the larger the relevant group, the poorer is the figural goodness. Good figures are devoid of information and require minimum cognitive processing

A final point should be made clear at this juncture. Again, good figures or good Gestalts impart little information so that they do not saddle the cognitive system with costly processing. Due to the minimal information in good Gestalts, such stimuli are easily perceived, recognized, learned, and recalled [18,19]. Could that be the reason why so many symbols of public systems are good figures? (see Figure 4).

## 6. Figural Goodness and Complexity

The measure of *R&R subset*, based on information theory, provides an objective definition for the critical Gestalt concept of figural goodness. A salient feature of good figures, in turn, is simplicity or noncomplexity. Could the vexing notion of complexity be defined in an objective fashion, too? Despite early efforts (e.g., [28,29], this property largely remained in the subjective realm. The breakthrough came with the advent of computational simulations in the last few decades. These programs centered around the notion of *algorithmic complexity* by way of capturing complexity quantitatively. Harnessing the computational power of complexity programs, Fitousi and Algom [2] have recently shown how to assess complexity in the Garner dot patterns (see Figure 1 again). It became possible to relate complexity and R&R subsets with respect to a given pattern—both measures are defined objectively. Notably, it became possible to assess the functional dependence of subjective figural goodness on objective complexity.

## 7. Algorithmic Complexity

Again, the basic intuition is that good figures are in some sense less random or complicated than bad figures. But in what sense? Kolmogorov’s theory of *algorithmic complexity* ([30] see also [31,32]) provided one answer: the complexity of a pattern—say a string of zeros and ones—is defined as the length of the shortest computer program that can (re)produce the string (see also [33]). It should be noted that Kolmogorov complexity is not computable and can only be approximated with respect to its upper bound. To this end, various compression algorithms have been developed. Thus, the popular *Lempel–Ziv Algorithm* (LZ; [34,35]) reads the string bit-by-bit; then, when it detects a unique substring, the program replaces it with a short code. The next time the algorithm meets this substring it uses this short code instead of the entire substring. Thus, the LZ algorithm provides a shorter result than the input string (especially when applied to very long input strings, often consisting of millions of bits). Another program is the *Effort-to-Compress Algorithm* (ETC; [36]). This algorithm identifies on each iteration the most frequent sequence and replaces it with a new symbol. The algorithm halts when the string becomes constant.

A final point should be noted with respect to all programs of algorithmic complexity. The output becomes compressed and efficient only if the input string includes patterns (e.g., palindromes, simple and mirror repetitions, and other regularities). Conversely, a completely random string—a patternless input—is not compressible. Randomness thus is isomorphic with incompressibility [37,38]. This provides a clue to the kinship with information theory and figural goodness. A random string imparts maximum information but, obviously, forms a poor stimulus.

## 8. Algorithmic Complexity of Two-Dimensional Patterns

Algorithmic complexity offers a promising avenue for studying figural goodness, yet applications are not free of problems. For one, the Garner patterns (Figure 1) and many pictures are two-dimensional stimuli, but the canonical Kolmogorov algorithms have concerned one-dimensional sequences [39,40]. Now, coding of two-dimensional images into one-dimensional digital sequences or vice versa is routine (think of GIF files), but the conversion is not trivial. For example, a one-dimensional string can be read from left to right and from right to left and give the exact same Kolmogorov value. In contrast, a two-dimensional pattern, like Garner’s dot patterns, can be read and coded in various ways, each coding producing a different value [2].

The assessment of two-dimensional complexity is important in many fields of science (e.g., [41,42]), yet, despite their serviceability, algorithms for complexity computation with two-dimensional patterns are in short supply (but see [39,43,44,45,46]). Several paths have been suggested for converting two-dimensional displays into one-dimensional sequences for which complexity is readily determined. One (e.g., [34,47]) chooses a single reading route of the two-dimensional stimulus and then computes the complexity of the resulting one-dimensional string as an estimate of the original two-dimensional complexity. This approach is suboptimal though because it does not capture all the intricate relations lurking in the two-dimensional pattern [39]. According to a second approach [2], the two-dimensional pattern is transformed to many one-dimensional sequences, each representing a specific reading of the two-dimensional stimulus. The distribution of values helps in recovering the information structure inherent to the two-dimensional pattern. A third solution is the development of dedicated two-dimensional complexity algorithms.

An intriguing measure of two-dimensional complexity has been developed by Zenil et al. [40]. The measure is based on Levin’s [48] coding theorem and ascribes a measure of complexity K(s) to a string(s) that is proportional to the probability of observing that pattern in halting two-dimensional Turing machines. A remarkable observation by Zenil et al. [40] was that the checkboard pattern (approximated by the pattern at the left of Figure 1) emerged as one of most frequent configurations. This high-probability pattern was marked by a low K(s) value, hence indicating simpleness or little complexity. Could this mean that good figures are not complex when the latter is assessed by measures of algorithmic complexity?

## 9. Dot-Pattern Gestalts Assessed by Algorithmic Complexity

The Garner dot patterns can be represented by the numerals 0 and 1, such that empty cells are denoted by 0, while cells occupied by a dot are represented by 1. For example, the left dot pattern in Figure 1 can be written as follows:
101010101

This 2D representation can then be encoded by its rows, starting from the top or from the bottom end and then from the left or right end. Or, the pattern can be encoded by columns, starting from the top or the bottom end and then from the left or from the right end. These readings result in eight ways of encoding a given pattern. For example, in the case of the left panel of Figure 1 as coded above, all eight readouts result in the same 1D string, irrespective of the way the pattern is encoded:
101010101

The observed pattern is remarkable. It can be read from left to right and from right to left with the result remaining invariant. This pattern is non-random, hence redundant. It is highly compressible. This is not the case with the pattern at the left of Figure 1. The 2D representation of the right pattern on Figure 1 is as follows.
100111010

In contrast to the previous pattern, encoding this poor pattern in the previous eight ways results in eight *different* 1D strings, as follows.
100111010010111100001111011010111001110011010011110010010011110010110011

The number of different strings generated by a given pattern is equal to the size of its R & R subset. For example, for the left pattern in Figure 1, both its R & R subset and the cardinality of its unique string equal 1, whereas for the right pattern in Figure 1, both equal 8. This result aligns with group theory. The number of different strings reflects the number of transformations under which the pattern remains invariant. For the good pattern, this amounts to eight, whereas for the poor pattern this amounts to one (identity).

## 10. Application of the LZ Measure of Complexity

How can this coding method help in recovering the complexity of the two-dimensional Garner patterns? A ready path is to apply any one of the compression algorithms developed for one-dimensional complexity to each of the different strings of zero and one. The average complexity can then serve as an approximation to the pattern’s two-dimensional complexity. This path was followed by Fitousi and Algom [2]. They calculated the average LZ complexity for all possible Garner dot patterns, separately for R&R subsets of 1, 4, and 8 (all possible values of the R&R set size), and they plotted the LZ complexity values as a function of the R&R subset size. Figure 5 presents their results.

A glimpse at Figure 5 reveals the presence of an association: patterns residing in larger R&R subsets are more complex as assessed by the mathematical measure of LZ complexity. However, the association is rather weak; the attenuation produced by severe constraints affects both variables. For LZ, the algorithm works best, almost exclusively, for large strings that comprise millions of bits. The Garner strings are extremely short. For R&R subsets, the variable has only three values (for these patterns). When viewed against these constraints, the appearance of an association is notable.

## 11. Application of the Subsymmetry Measure of Complexity

Another way of computing the two-dimensional complexity needed for the Garner dot patterns is by the measure of subsymmetry [28] and its extension to two-dimensional patterns [29,49]. Subsymmetry is the subset of neighboring elements within the stimulus string that have mirror symmetry. Nagaraj and Balasubramanian [36] defined a normalized measure of subsymmetries for unidimensional sequences of 0 and 1 as follows.(3a)$$\mathrm{SubSym}(x)=1-\frac{subsymmetries(x)}{subsymmetries(Z)}$$(3b)$$=1-\frac{subsymmetries(x)}{\left(\frac{n(n-1)}{2}\right)}$$
where x is the length n of the input, and Z is the uniform, all-zero or all-one, sequence of that length n. The function computes the number of subsymmetries contained in the sequence. Note that the uniform sequence has n(n − 1)/2 subsymmetries, the largest possible number. To provide a normalized measure of complexity, the ratio in Equation (3a) is subtracted from 1 in Equation (3b). This normalized measure is bounded between 0 (least complex) and 1 (most complex). Less formally, the larger the number of subsymmetries contained in the sequence, the less complex it is. Good figures translate into structured strings with many subsymmetries; hence, they are not complex by this measure. By contrast, a string devoid of any subsymmetry has maximal complexity.

Again, the measure developed by Nagaraj and Balasubramanian [36] is one-dimensional, but the same routine performed with the LZ complexity can be repeated with the subsymmetry measure. Average subsymmetry can be calculated over all one-dimensional strings issuing from a given dot pattern. Fitousi and Algom [2] performed this for all the Garner dot patterns and plotted the average complexity against the R&R subset size. The results were very similar to those obtained with the LZ measure: a weak association.

However, it is possible to derive a measure harnessing the local symmetry or subsymmetry idea that better approximates the complexity of two-dimensional displays. Following Toussaint et al. [49], Fitousi and Algom [2] developed a measure, SubSym, that considers the separate contributions of vertical (vSub), horizontal (hSub), right-diagonal (rdSub), and left-diagonal (ldSub) subsymmetries. The measure is as follows.(4a)$$\mathrm{SubSym}(x)=1-\frac{\sum_{i=1}^{n}\sum_{j=1}^{n}\sum_{k=1}^{k=2}L_{ijk}SubSym(x)}{\sum_{i=1}^{n}\sum_{j=1}^{n}\sum_{k=1}^{k=2}L_{ijk}SubSym(Z)}$$(4b)$$=1-\frac{TotSub(x)}{TotSub(Z)}$$
where TotSub is the sum of all subsymmetries, Z is the uniform pattern, i and j are the number of rows and columns, respectively, k is the number of diagonals (2), and L is the length of the attendant string. Note that the various subsymmetries included in SubSym apply directly to the spatial patterns themselves.

Fitousi and Algom [2] calculated the SubSym complexity for each of Garner’s dot patterns. They then plotted the *subjective* rating of figural goodness for each pattern (as reported in [27]) against its objective complexity (Figure 6). Fitousi and Algom [2] found an appreciable correlation of −0.54 between the two, providing the clearest demonstration to date that perceptually good figures are less complex in an objective mathematical sense.

The currently available tools of algorithmic complexity enable, for the first time, to define the hitherto subjective Gestalt notions of goodness, simplicity, pattern-ness, or order in an objective rigorous way. The displays shown in Figure 5 and Figure 6 converge on the same result: people’s impression of the goodness of a pattern is predicted by the pattern’s complexity, defined in a rigorous mathematical manner. Thus, a good pattern in perception also means that it resides in a small set of subjectively similar patterns (Figure 5), and good patterns to perception are objectively less complex than poor patterns. The poorer the pattern to subjective impression, the more complex it is (Figure 6).

## 12. Conclusions

Shannon’s [1] formulation of “information theory” made a huge fuss among students of perception of the day. It still does. However, the application of information measures to the perception of form proved challenging. Early attempts ran into a major stumbling block: the lack of a psychological counterpart to the informational primitive of set. Once this crucial impediment was removed, it cleared the way for a wealth of creative developments in perception. A current culmination is the application of algorithmic complexity to Gestalt-type stimuli and concepts. The present analysis is an inchoate first foray into this promising domain.

## Figures and Tables

**Figure 1 entropy-27-00434-f001:**
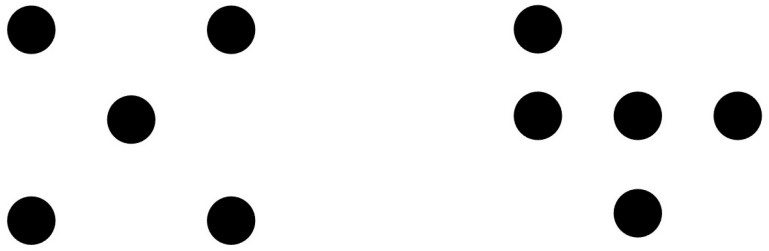
Two dot patterns defined by their R & R subsets and their symmetry groups. (**Left panel**): Size of the R & R subset = 1; size of symmetry group = 8 [I, V, H, L, R, 90°, 180°, and 270° rotations]. (**Right panel**): Size of the R & R subset = 8; similarity group = 1 [I]. I: identity; V: vertical reflection; H: horizontal reflection; L: left-diagonal reflection; R: right-diagonal reflection. Based on [2,19].

**Figure 2 entropy-27-00434-f002:**
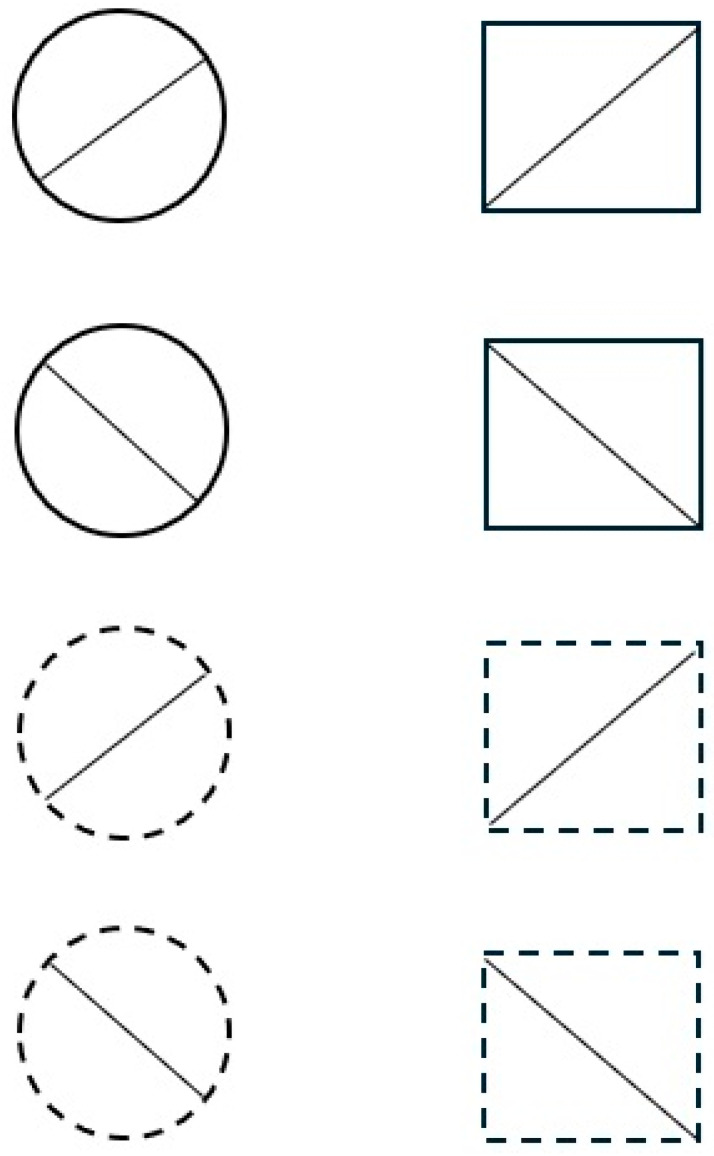
Total set of 8 stimuli formed by crossing the binary values of 3 dimensions: form (circle, square) × slope (upwards, downwards) x line (solid, broken) (based on [21]).

**Figure 3 entropy-27-00434-f003:**
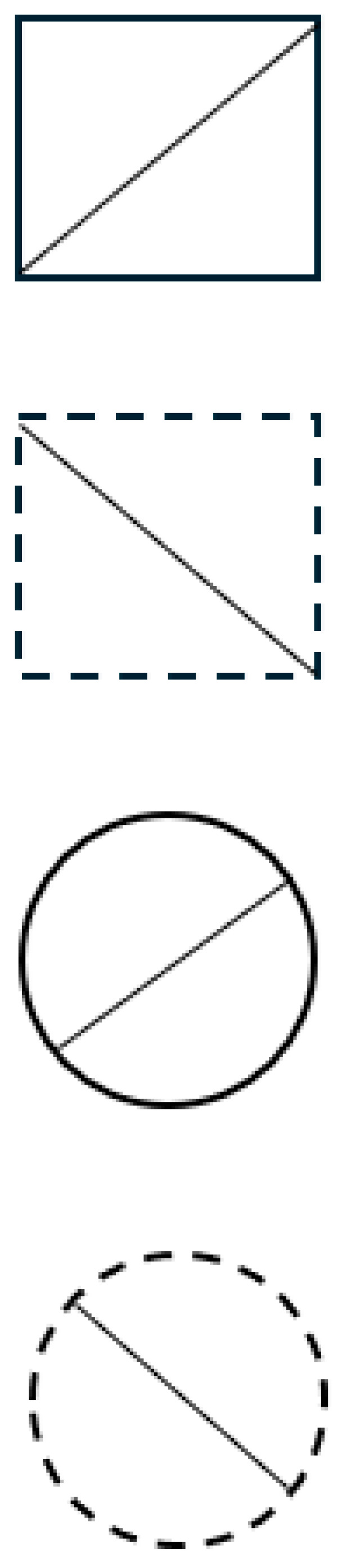
A subset of 4 stimuli drawn from the total set of Figure 2. Notice the correlation between line and slope: with solid lines, the slope is upward, whereas with broken lines, the slope is downward. All subsets must contain redundancy or correlation.

**Figure 4 entropy-27-00434-f004:**
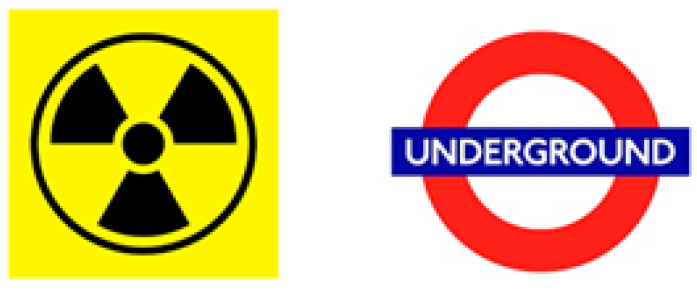
A pair of well known images of public institutions and systems. Notice the appreciable amounts of symmetry, hence little information, in each symbol. They are easily perceived, recognized, and remembered.

**Figure 5 entropy-27-00434-f005:**
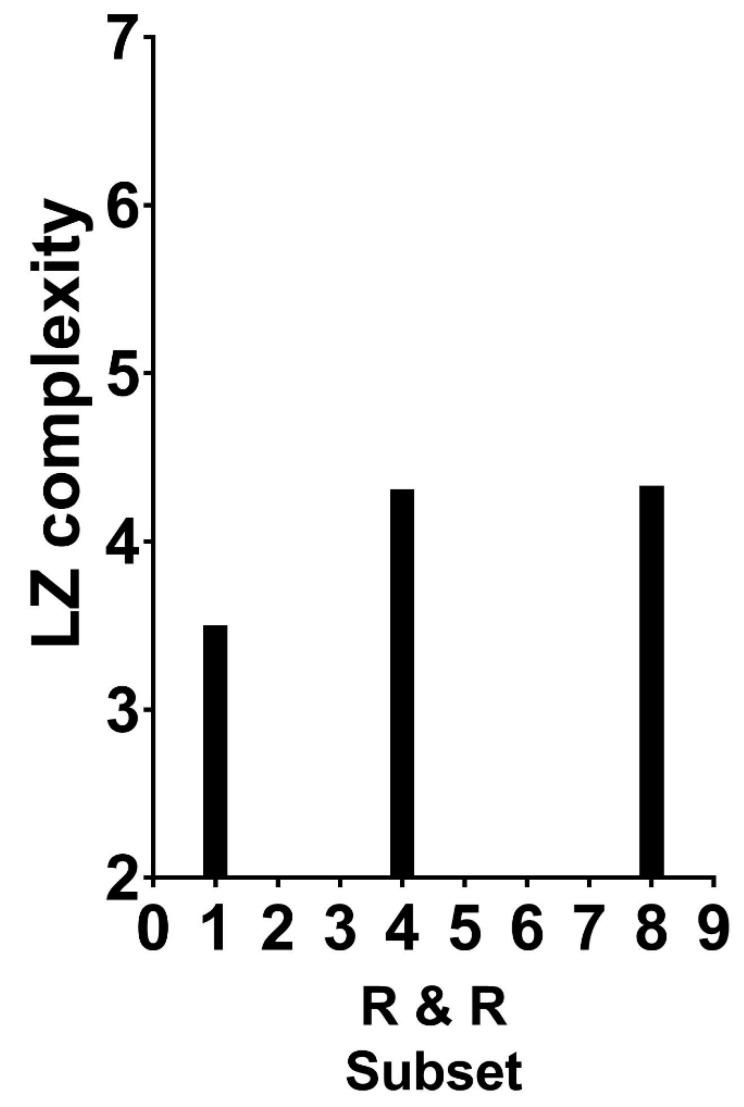
Average LZ complexity as a function of the R & R subset.

**Figure 6 entropy-27-00434-f006:**
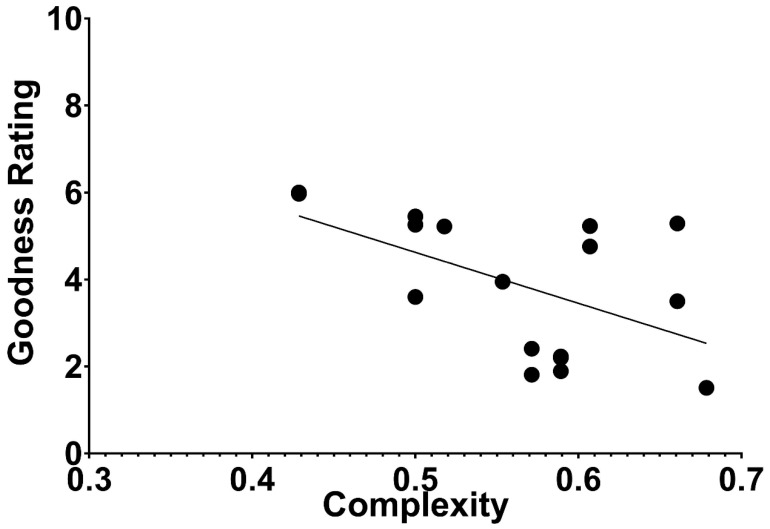
Judged figural goodness of a pattern as a function of the pattern’s complexity as assessed by SubSym. The ratings of figural goodness were taken from [27]. The Pearson correlation between the two variables is −0.54.

## Data Availability

The data presented in this study are available on request from the corresponding author.

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
