# Peer review of "Information Theory in Perception of Form: From Gestalt to Algorithmic Complexity"

_entropy, 2025, doi:10.3390/e27040434_

Round 1

Reviewer 1 Report

Comments and Suggestions for Authors

Thank you for the opportunity to read this manuscript, which provides an insightful and interesting overview of the links between information theory and beauty (well, maybe not quite beauty, but close). My recommendation is that this manuscript should be published after minor revision. My suggestions for revision are a few things which the authors might consider adding to the manuscript. I can appreciate that adding things can also blunt the focus of the writing, so I will understand if the authors do not wish to make these additions:

  1. Garner’s proposal to define a comparison set using the equivalence class defined by some transforming relationships is brilliant. But it also always struck me as a punting the question a little. Which relationships? Using reflection and rotation makes sense, but there are many other relations on the plane which could have been chosen, were not chosen, and would lead to very different equivalence classes (comparison sets). This probably isn’t a problem in 2D, but surely becomes a problem with judging solid objects in 3D or videos which add the dimension of time. I wonder if the authors have ideas about how this could be tackled?
  2. One of the most influential and popular theoretical frameworks of modern psychology is the “free energy principle”, popularised by Karl Friston. Given it’s huge impact on psychology and psychiatry, I expected to read a little about it’s close links with information theory. I realise it is not directly related to judging perceptual goodness, but there still is probably still a reasonable link. For devotees of the free energy principle, the role of the brain is to find “good” inferences about the state of the world, based on noisy perceptual inputs. The “good” here is defined by free energy, which is equivalent to Shannon information, under reasonable assumptions.
  3. Similar to #2, but maybe a bit more tenuous! I like to think of myself when developing scientific theory as hunting for “good” theoretical explanations. What is “good” about an explanation is very important to understand, but not easy to define. The link to the current work is that many scientific researchers adopt Bayesian inference methods for theory testing and model selection. Bayesian inference is tightly linked to information theory. This suggests that maybe what Bayesian inference is doing in model selection, while couched in terms of maximising posterior probabilities, is also maximising some “goodness” of the theory. I admit this is pretty speculative stuff, but maybe the authors have clearer ideas about it than me.

Author Response

  1. Garner’s proposal to define a comparison set using the equivalence class defined by some transforming relationships is brilliant. But it also always struck me as a punting the question a little. Which relationships? Using reflection and rotation makes sense, but there are many other relations on the plane which could have been chosen, were not chosen, and would lead to very different equivalence classes (comparison sets). This probably isn’t a problem in 2D, but surely becomes a problem with judging solid objects in 3D or videos which add the dimension of time. I wonder if the authors have ideas about how this could be tackled?

Thank you much for this thoughtful comment. We do think about this issue, but it requires another paper-wide treatment. So, we are leaving the current text as is.

2.  One of the most influential and popular theoretical frameworks of modern psychology is the “free energy principle”, popularised by Karl Friston. Given it’s huge impact on psychology and psychiatry, I expected to read a little about it’s close links with information theory. I realise it is not directly related to judging perceptual goodness, but there still is probably still a reasonable link. For devotees of the free energy principle, the role of the brain is to find “good” inferences about the state of the world, based on noisy perceptual inputs. The “good” here is defined by free energy, which is equivalent to Shannon information, under reasonable assumptions.

Again, we fully respect your insightful comment. Your point is brilliant. It is well taken especially we are aware and admire Friston's work. At this point though the idea requires more thinking and work -- we do plan for the future.

 3. Similar to #2, but maybe a bit more tenuous! I like to think of myself when developing scientific theory as hunting for “good” theoretical explanations. What is “good” about an explanation is very important to understand, but not easy to define. The link to the current work is that many scientific researchers adopt Bayesian inference methods for theory testing and model selection. Bayesian inference is tightly linked to information theory. This suggests that maybe what Bayesian inference is doing in model selection, while couched in terms of maximising posterior probabilities, is also maximising some “goodness” of the theory. I admit this is pretty speculative stuff, but maybe the authors have clearer ideas about it than me.

A good point again, we certainly see the possible connection to Bayesian inference. In this paper we pursued goodness of perception. Goodness of theories is another great idea to pursue next.

Thank you very much for the truly insightful comments! Certainly, our work next is cut out for us.

Sincerely,

Daniel Algom

Reviewer 2 Report

Comments and Suggestions for Authors

I think the work is relevant and well presented. The application of Gestalt principles in automatic image recognition with good cognitive patterns is certainly promising. We are only at the beginning but the work opens up very interesting perspectives.

Author Response

I think the work is relevant and well presented. The application of Gestalt principles in automatic image recognition with good cognitive patterns is certainly promising. We are only at the beginning but the work opens up very interesting perspectives.

Thank you much for the positive evaluation. We agree, we are certainly at the beginning of uncovering complexity and cognitive patterns.

Sincerely,

Daniel Algom